# Associations Between DNA Repair Gene Polymorphisms and Breast Cancer Histopathological Subtypes: A Preliminary Study

**DOI:** 10.3390/jcm14113764

**Published:** 2025-05-27

**Authors:** Claudiu Ioan Filip, Andreea Cătană, Lorin-Manuel Pîrlog, Andrada-Adelaida Pătrășcanu, Mariela Sanda Militaru, Irina Iordănescu, George Călin Dindelegan

**Affiliations:** 1Department of Plastic Surgery and Burn Unit, Emergency District Hospital, 400535 Cluj-Napoca, Romania; filip_claudiu2000@yahoo.com (C.I.F.); george.dindelegan@umfcluj.ro (G.C.D.); 2First Surgical Clinic, Faculty of Medicine, University of Medicine and Pharmacy “Iuliu Hatieganu”, 400012 Cluj-Napoca, Romania; 3Department of Molecular Sciences, Faculty of Medicine, University of Medicine and Pharmacy “Iuliu Hațieganu”, 400012 Cluj-Napoca, Romania; catanaandreea@gmail.com (A.C.); adelaidapatrascanu@gmail.com (A.-A.P.); mariela.militaru@reginamaria.ro (M.S.M.); 4Department of Oncogenetics, Institute of Oncology, “Prof. Dr. I. Chiricuță”, 400015 Cluj-Napoca, Romania; 5Regional Laboratory Cluj-Napoca, Department of Medical Genetics, Regina Maria Health Network, 400363 Cluj-Napoca, Romania; 6Department of Medical Genetics, Genetic Centre Laboratory, Regina Maria Health Network, 011376 Bucharest, Romania; irina.iordanescu@reginamaria.ro

**Keywords:** CHEK2, XRCC1, XPD, triple-negative breast cancer, nucleotide excision repair pathway, base excision repair pathway, DNA repair pathway

## Abstract

**Introduction**: This study investigates the distribution and interaction of three polymorphisms—XRCC1 (rs1799782), CHEK2 (rs17879961), and XPD (rs238406)—in Romanian breast cancer patients, aiming to understand their association with histopathological subtypes, age, and BMI. **Materials and Methods**: This retrospective study analyzed 36 breast cancer patients from a Romanian clinic (2020–2024) with complete genetic data for XRCC1 (rs1799782), CHEK2 (rs17879961), and XPD (rs238406). The patients had invasive, non-metastatic breast cancer and no history of other cancers. Statistical analysis with Jamovi included descriptive stats, McNemar’s test for genotype associations, and multinomial logistic regression to explore links between variants, age, BMI, and tumor subtypes. **Results**: McNemar tests showed no significant association between XRCC1 and CHEK2 (*p* = 0.180), nor between XRCC1 and XPD (*p* = 0.03) or XPD and CHEK2 (*p* = 0.049) after applying the Bonferroni correction (α = 0.0167), indicating no statistically significant genetic dependency among these variants. A multinomial logistic regression model found that genetic variants, BMI, and age significantly predicted breast cancer subtypes, particularly CDI TNB. All predictors remained significant in the comparisons of CDI TNB vs. CDI LB/CDI LA. Notably, these associations remained unchanged even after applying the Bonferroni correction (α = 0.0021), confirming the robustness of the findings. **Conclusions**: This study identifies significant associations between XRCC1, CHEK2, and XPD gene variants and CDI TNB, suggesting a role of DNA repair deficiencies in its pathogenesis. Protective genotypes were under-represented in TNB cases. Limited links with luminal subtypes highlight TNB’s distinct genetic profile. Results support further research on these polymorphisms as markers for TNB risk and precision treatment.

## 1. Introduction

Breast cancer (BC) remains one of the most prevalent and challenging health concerns globally, accounting for approximately 22.9% of all cancers in women and representing one of the leading causes of cancer-related mortality in this population group [1,2,3,4]. Notably, 5–10% of all breast cancer cases are attributed to inherited genetic predisposition [3]. While BRCA1 and BRCA2 are the most well-known high-penetrance genes associated with a substantial increase in breast cancer risk, there are also more common but lower-penetrance genes, such as CHEK2, that contribute to breast cancer susceptibility and are increasingly being considered in risk assessment [4].

### 1.1. Histopathological Types of Breast Cancer 

A key aspect of modern BC management is its molecular classification, which reflects the substantial heterogeneity of the disease and significantly influences treatment decisions and prognostic evaluation [5]. The molecular categorization of BC is based on the evaluation of four clinically standardized biomarkers: estrogen receptor (ER), progesterone receptor (PR), human epidermal growth factor receptor 2 (HER2), and the proliferation marker Ki67 [1,6,7]. Based on the presence or absence of these markers, breast cancer is commonly classified into at least four intrinsic subtypes, Luminal A (LA), Luminal B (LB), HER2-enriched, and triple-negative breast cancer (TNB), each associated with distinct clinical outcomes [1,6].

Luminal subtypes are generally ER-positive and are further divided into LA and LB. LA tumors typically exhibit high expression of ER-related genes, low Ki67 levels (indicating low proliferation), and positive PR status, resulting in a relatively favorable prognosis [8,9,10]. In contrast, LB tumors show lower expression of ER-related genes, higher proliferation (as indicated by elevated Ki67), and may have lower or negative PR expression, contributing to a poorer prognosis [9,10]. Clinically, the surrogate classification of luminal subtypes is performed using immunohistochemical analysis for ER, PR, HER2, and Ki67, which guide therapeutic decisions [10]. Moreover, L B can be further divided into LB-like HER2-negative and LB-like HER2-positive [6].

TNB lacks the expression of ER, PR, and HER2, making it difficult to treat with hormone or HER2-targeted therapies. TNB is associated with a higher risk of early relapse, aggressive metastatic behavior, and poorer overall prognosis [11]. The 2013 St. Gallen International Breast Cancer Conference further emphasized this molecular classification as a framework for personalized treatment strategies, aiming to reflect the biological diversity of breast cancer and improve patient outcomes [12].

### 1.2. DNA Repair Pathways in Breast Cancer: Roles of XRCC1, XPD, and CHEK2

Two major factors that contribute to the development and spread of BC are genomic instability and DNA damage. Double-strand breaks (DSBs) are one of the most harmful types of DNA lesions [13,14]. Cells use two main DSB repair pathways to maintain genomic integrity: non-homologous end joining (NHEJ), a more error-prone but cycle-independent method, and homology-directed repair (HDR), which is error-free and active during the S/G2 phases [15]. Together, these mechanisms preserve the stability of the DNA.

The nucleotide excision repair (NER) pathway is essential for repairing large DNA lesions brought on by environmental mutagens in addition to DSB repair [16]. A key component of NER, the XPD gene encodes a helicase that unwinds DNA and detects damage, including thymine dimers and DNA adducts [16]. NER functions through two sub-pathways: transcription-coupled NER (TC-NER) and global-genome NER (GG-NER). They differ in how they identify damage but have a similar repair process [17,18,19]. Proteins such as TFIIH (including XPB/XPD), XPA, and RPA come together to process the lesion after damage is detected. Following the removal of the damaged DNA by endonucleases ERCC1/XPF and XPG, POLδ/ε, PCNA, and ligase aid in its resynthesis and ligation [17,20,21].

Base excision repair (BER) is used to treat tiny base lesions that do not deform helices. Together with POLβ and LIG3, XRCC1 forms a scaffold in short-patch BER, controlling PARP1 activity and promoting effective single-strand break repair [17]. In the 5′-Gap sub-pathway, where ERCC1/XPF, RECQ1, and PARP1 are active, XRCC1 may also indirectly control PARP1 toxicity in long-patch BER [22].

The susceptibility to breast cancer is greatly influenced by polymorphisms in these important DNA repair genes. For example, helicase function may be compromised by the XPD rs238406 variation, which would lower NER performance and raise the risk of mutagenesis. Like this, the XRCC1 rs1799782 mutation has been linked to faulty BER, which results in increased genomic instability and oxidative DNA damage buildup. These are indicators of cancer progression, especially in aggressive subtypes like TNB [17,20,21,22].

Furthermore, research shows that ERCC1/XPF is essential for NER as well as for processing oxidative DNA lesions caused by reactive oxygen species (ROS), such as 3′-phosphoglycolates (3′-PG). The crucial function that DNA repair plays in preventing cancer is highlighted by the fact that deficiencies in this pathway result in increased sensitivity to ROS and decreased cell survival [23].

Simultaneously, a serine/threonine kinase encoded by the CHEK2 gene phosphorylates downstream effectors in cell cycle checkpoints in response to DNA damage [24]. Carriers of germline mutations in CHEK2 have an estimated 25% to 30% lifetime probability of developing breast cancer, which is associated with a moderate risk [25,26]. Not all carriers of CHEK2 mutations develop cancer, even though they are somewhat frequent (0.3–1.6% in the general population, up to 5.7% in individuals with a family history) [27]. This variation raises the possibility that other genetic modifiers, such as variations in DNA repair genes like XRCC1 and XPD, could increase the risk of cancer in people with CHEK2 [28,29].

Although CHEK2’s involvement in cancer susceptibility is becoming increasingly clearer, nothing is known about how DNA repair gene polymorphisms combine to influence this risk. Precision medicine strategies for screening, prevention, and therapy may be made possible by the integration of polygenic risk scores, such as variations in XRCC1, XPD, and CHEK2, as genome sequencing technologies develop [30,31,32].

Avoiding cancer critically depends on the integrity of DNA repair mechanisms. By impairing the repair of crucial DNA lesions, functional polymorphisms in XRCC1, XPD, and CHEK2 contribute to the development and severity of breast cancer. Their research advances our knowledge of the pathophysiology of breast cancer and aids in the creation of individualized treatment and diagnostic strategies.

### 1.3. Study Objectives and Scope

This study was conceived in response to this knowledge gap. Although several studies have individually explored the roles of XRCC1 (rs1799782), CHEK2 (rs17879961), and XPD (rs238406) in cancer risk, few have evaluated their combined interaction or distribution across breast cancer subtypes in a clinically annotated patient population.

The aim of this study was to conduct a preliminary, retrospective analysis of the distribution and inter-relationship of XRCC1, CHEK2, and XPD polymorphisms in a Romanian cohort of breast cancer patients, with a specific focus on their potential association with histopathological subtypes, age, and BMI. By leveraging existing genetic testing data from a real-world clinical setting, we sought to determine whether patterns of genetic variation could help explain some of the observed heterogeneity in breast cancer presentation.

The objective was twofold: first, to assess whether there is a statistical association or co-occurrence between these key polymorphisms in the same individuals; and second, to evaluate their predictive relevance for distinguishing among major histopathological classifications, which remains a therapeutic challenge.

The scope of this research is exploratory and hypothesis-generating. While limited by sample size, this study contributes valuable early data that could inform the design of larger, prospective investigations. In doing so, it opens the possibility of integrating moderate-risk genetic variants into future molecular stratification strategies—ultimately aiming to support precision oncology approaches in breast cancer management.

## 2. Materials and Methods

### 2.1. Patient Selection and Study Design

This study was designed as a retrospective analysis of a breast cancer cohort treated at the Regina Maria Private Health Network in Cluj-Napoca, Romania. A total of 745 female patients diagnosed with breast cancer between January 2020 and December 2024 were initially identified through institutional medical records.

Among these, a subgroup of 36 patients was selected for inclusion based on the availability of complete genetic testing data. These patients had previously undergone molecular analysis for polymorphisms in DNA repair genes as part of their diagnostic or prognostic evaluation, routinely integrated within the clinical genetic services offered at Regina Maria. The availability of existing genotyping results enabled the retrospective exploration of potential associations between specific variants and histopathological characteristics.

Inclusion criteria for this preliminary study were as follows: (1) a confirmed histopathological diagnosis of invasive breast cancer; (2) no evidence of metastatic disease at the time of diagnosis; (3) absence of other primary cancers in the patient’s medical history; (4) documented genotyping results for XRCC1 (rs1799782), CHEK2 (rs17879961), and XPD (rs238406) available in the medical record (see Figure 1).

This approach allowed for the use of high-quality, clinically integrated genetic data in a research context, minimizing the need for additional patient interventions. The selected sample was used to evaluate the distribution and interplay of key DNA repair gene polymorphisms in relation to patient age, BMI, and histopathological breast cancer subtype.

### 2.2. Genetic Testing

Sequencing data from the Illumina platform (Illumina Inc., San Diego, CA, USA) that were aligned to the human reference genome (GRCh37/hg19) were used for genetic testing. Using specialized software tools and the gnomAD and ClinVar databases, variants were examined. GATK (version 4.3.0.0, Broad Institute, Cambridge, MA, USA) was used for variant calling, and VarSeq (version 2.4.0, Golden Helix, Bozeman, MT, USA) and Alamut Visual (version 1.11, SOPHiA GENETICS, Rolle, Switzerland) were used for annotation and interpretation. Using ExomeDepth (version 1.1.15, University of Cambridge, Cambridge, UK), copy number variations (CNVs) were found. A clinical team assessed pathogenicity in accordance with ACMG recommendations, and Sanger sequencing with the ProDye^®^ Terminator Sequencing System (Promega Corporation, Madison, WI, USA) was used to confirm significant findings, as well as via Eurofins Genomics’ Sanger sequencing services (Ebersberg, Germany). XRCC1 (rs1799782) and XPD (rs238406) were genotyped using the (M-PCR-RFLP) multiplex polymerase chain reaction–restriction fragment length polymorphism technique [33,34].

### 2.3. Applied Statistical Methods

All statistical analyses were performed using Jamovi software (version 2.6.17, The Jamovi Project, Sydney, Australia). Continuous variables such as age and body mass index (BMI) were summarized using descriptive statistics, including mean, standard error (SE), standard deviation (SD), median, and range. The normality of distribution was assessed using the Shapiro–Wilk test.

Associations between categorical variables, particularly genotype distributions of XRCC1 (rs1799782), CHEK2 (rs17879961), and XPD (rs238406), were assessed using paired contingency tables analyzed via the McNemar test. This test was chosen due to the within-subject nature of the data, as everyone contributed multiple polymorphism measurements.

To evaluate the predictive relationship between genetic variants and histopathological classification, a multinomial logistic regression model was applied. Histopathological subtypes—CDI LB, CDI TNB, and CDI LA—were entered as the dependent variable, with CDI LB set as the reference category. Independent variables included the three genotyped polymorphisms, age, and BMI. The goodness-of-fit of the model was evaluated using deviance, the Akaike Information Criterion (AIC), and McFadden’s pseudo-R^2^.

Statistical significance was set at *p* < 0.05 for all analyses. As this study was exploratory in nature, no correction for multiple comparisons was applied; however, the results are interpreted as hypothesis-generating and require confirmation in larger, independent cohorts.

## 3. Results

The results of this preliminary study, including descriptive statistics, bivariate associations between genetic polymorphisms, and multivariate modeling through multinomial logistic regression, are presented in the following subsections. The analysis aimed to explore the distribution of XRCC1 (rs1799782), CHEK2 (rs17879961), and XPD (rs238406) genotypes in a cohort of female breast cancer patients and their potential association with clinical characteristics and histopathological subtypes.

### 3.1. Descriptive Statistical Analysis

Table 1 presents the descriptive statistics for the continuous variables included in this study—age and body mass index (BMI)—within the breast cancer cohort. Both variables are summarized with their means, standard errors (SEs), confidence intervals (CIs), medians, standard deviations (SDs), and ranges. The Shapiro–Wilk test was used to assess the normality of age and BMI distributions in the study cohort.

### 3.2. Genetic Dependency Analysis

In this section, we present the results of three McNemar tests conducted to assess the pairwise associations between the genotypes of XRCC1 (rs1799782), CHEK2 (rs17879961), and XPD (rs238406). Given the multiple statistical comparisons performed, we aimed to control the risk of Type I errors (false positives) by applying the Bonferroni correction. The adjusted significance threshold (α) was calculated as 0.05/3 = 0.0167. Therefore, in the interpretation of results, we considered associations to be statistically significant only if the raw *p*-values from the McNemar tests were below this corrected threshold.

A McNemar test was conducted to assess the association between the XRCC1 (rs1799782) and CHEK2 (rs17879961) genotypes. The result was not statistically significant, χ^2^ = 4.89, *p* = 0.180, indicating no evidence of dependency between the distributions of these two polymorphisms in the studied cohort. Considering the Bonferroni-adjusted significance threshold (α = 0.0167) applied to account for multiple comparisons, this result remains non-significant. Table 2 represents the contingency table between CHEK2 (rs17879961) and XRCC1 (rs.1799782) polymorphism.

A McNemar test was conducted to assess the association between the XPD (rs238406) and XRCC1 (rs.1799782) genotypes. The result was statistically significant, χ^2^ = 8.93, *p* = 0.03, indicating evidence of dependency between the distributions of these two polymorphisms in the studied cohort. However, when applying the Bonferroni-adjusted significance threshold (α = 0.0167) to account for multiple comparisons, this result did not reach the corrected level of significance. Table 3 represents the contingency table between the XPD (rs238406) and XRCC1 (rs.1799782) polymorphisms.

A McNemar test was conducted to assess the association between the XPD (rs238406) and CHEK2 (rs17879961) genotypes. The result was statistically significant, χ^2^ = 7.85, *p* = 0.049, indicating evidence of dependency between the distributions of these two polymorphisms in the studied cohort. However, when applying the Bonferroni-adjusted significance threshold (α = 0.0167) to account for multiple comparisons, this result did not reach the corrected level of significance. Table 4 represents the contingency table between the XPD (rs238406) and CHEK2 (rs17879961) polymorphisms.

### 3.3. Multinomial Analysis of Genetic Markers in Breast Cancer Histopathological Subtypes

In this section, we present the results of a multinomial logistic regression analysis examining the association between genetic polymorphisms and histopathological subtypes of breast cancer. The analysis compares three subtypes—CDI TNB, CDI LA, and CDI LB—through pairwise comparisons: CDI TNB vs. CDI LB, CDI LA vs. CDI LB, and CDI TNB vs. CDI LA. For each comparison, we evaluated the effects of the predictors XRCC1 (rs1799782) [genotype contrasts: AA vs. AT, TT vs. AT], XPD (rs238406) [AA vs. CA, CC vs. CA], CHEK2 (rs17879961) [CC vs. TC, TT vs. TC], BMI, and age, yielding a total of 24 statistical tests. To control for the risk of Type I errors due to multiple comparisons, we applied a Bonferroni correction. Therefore, in interpreting the results, only *p*-values below 0.0021 are considered statistically significant after correction.

The multinomial logistic regression model used to evaluate the association between genetic polymorphisms and histopathological subtypes of breast cancer demonstrated a good fit, with a deviance of 29.9, an AIC of 65.9, and a McFadden’s pseudo-R^2^ of 0.562, indicating that approximately 56.2% of the variance in histopathological classification could be explained by the predictors in the model (Table 5).

## 4. Discussion

This preliminary study investigated the distribution and potential associations between three DNA repair gene polymorphisms—XRCC1 (rs1799782), CHEK2 (rs17879961), and XPD (rs238406)—in a cohort of female patients diagnosed with breast cancer. In addition to analyzing genotypic interactions, we evaluated their relationship with histopathological subtypes and clinical parameters such as age and body mass index (BMI). Despite the modest sample size (N = 36), there emerged several important patterns that may contribute to the growing body of literature on breast cancer genetics and molecular heterogeneity.

### 4.1. Genetic and Clinical Characteristics of the Study Cohort and Interpolymorphism Associations

This section presents key descriptive and inferential statistical analyses that contextualize the genetic and clinical data used in the breast cancer subtype analysis. These results are essential for understanding the characteristics of the study cohort and the inter-relationships between the examined genetic polymorphisms: XRCC1 (rs1799782), XPD (rs238406), and CHEK2 (rs17879961). The data also provide insights into whether these polymorphisms may occur independently or in combination patterns suggestive of functional or inherited associations.

The descriptive statistics in Table 1 summarize the age and body mass index (BMI) of the breast cancer cohort. The mean age was 62.1 years (SE = 1.84), with a median of 63.0 and a standard deviation (SD) of 11.04, suggesting a mature cohort with a relatively tight age distribution. The minimum age recorded was 38, and the maximum was 82, highlighting that the sample encompassed both pre- and postmenopausal women, although the majority were older. The Shapiro–Wilk test for normality showed no significant deviation from a normal distribution (W = 0.980, *p* = 0.735), supporting the suitability of parametric statistical methods for age-related analysis.

BMI followed a similar pattern, with a mean of 28.7 (SE = 1.02), suggesting an overweight population on average. The median BMI was 28.5, with a standard deviation of 6.10, indicating moderate variability in body composition. The Shapiro–Wilk test again confirmed a normal distribution (W = 0.970, *p* = 0.418). These results establish the clinical profile of the cohort and validate the use of BMI and age as continuous variables in downstream analyses such as logistic regression.

To explore potential genetic associations, a series of McNemar tests and contingency tables (Table 2, Table 3 and Table 4) were used to examine pairwise relationships between the three polymorphisms. These tests are vital in evaluating whether the observed genotype frequencies suggest non-random associations—i.e., linkage disequilibrium or functional co-selection—within this cohort.

The first McNemar test assessed the association between XRCC1 (rs1799782) and CHEK2 (rs17879961). The result was not statistically significant (χ^2^ = 4.89, *p* = 0.180), as shown in Table 2. Considering the Bonferroni-adjusted significance threshold (α = 0.0167) applied to account for multiple comparisons, this result remains non-significant. This suggests that the genotype distribution of XRCC1 is independent of the distribution of CHEK2 in this population. In other words, there is no evidence that carrying a particular variant of XRCC1 increases or decreases the likelihood of carrying a specific CHEK2 genotype. The independence of these polymorphisms may indicate that they reside on different chromosomal regions or are not subject to shared evolutionary pressures in this sample.

In contrast, the second McNemar test revealed an apparent statistically significant association between XRCC1 (rs1799782) and XPD (rs238406), with a χ^2^ value of 8.93 and *p* = 0.030, as shown in Table 3. This result initially suggests a dependency between the distributions of these two polymorphisms, implying that certain XRCC1 genotypes may co-occur more frequently with specific XPD variants than expected by chance. However, when applying the Bonferroni-adjusted significance threshold (α = 0.0167) to account for multiple comparisons, this association did not reach the corrected level of statistical significance. While the uncorrected result hints at a possible biological link, it should be interpreted with caution. Both XRCC1 and XPD are involved in DNA repair pathways—XRCC1 in base excision repair and XPD in nucleotide excision repair—so any observed association could reflect coordinated selection pressures or functional interactions. It remains plausible that these genes participate in shared repair mechanisms, and particular genotype combinations might exert synergistic effects, either protective or pathogenic, in breast tissue. Nonetheless, further studies with larger sample sizes are needed to confirm this potential relationship.

A similar pattern was observed in the third McNemar test, which evaluated the relationship between XPD (rs238406) and CHEK2 (rs17879961). The test yielded a statistically significant result (χ^2^ = 7.85, *p* = 0.049), indicating a potential dependency between the distributions of these two polymorphisms (Table 4). However, when applying the Bonferroni-adjusted significance threshold (α = 0.0167) to account for multiple comparisons, this association did not meet the corrected level of statistical significance and should therefore be interpreted with caution. Despite this, the uncorrected result may still hint at a biological relationship, but it should be interpreted with caution. CHEK2 is a key regulator of cell cycle control and DNA damage response, while XPD contributes to the repair of bulky DNA lesions via nucleotide excision repair. Their potential interaction may influence critical cellular processes such as repair efficiency or the decision between DNA repair and apoptosis. The co-occurrence of certain genotypes might alter this balance, potentially affecting breast cancer susceptibility or progression. Further investigation in larger and independent cohorts is needed to validate this possible association.

Overall, these findings suggest that the XRCC1 and CHEK2 genotypes segregate independently, while the combinations of XRCC1 with XPD and XPD with CHEK2 exhibit associations that are statistically significant at the uncorrected level but do not remain significant after applying the Bonferroni correction (α = 0.0167). Although these results hint at potential non-random relationships between specific gene pairs, none of the associations withstand adjustment for multiple testing. This highlights the need for cautious interpretation and underscores the importance of rigorous statistical control when exploring gene–gene interactions. Nonetheless, the observed trends may still reflect underlying biological relationships, such as coordinated involvement in DNA repair pathways or evolutionary pressures favoring certain genotype combinations. From a methodological standpoint, these findings suggest that potential gene–gene interdependencies should be considered in multivariate models, as they may influence or amplify associations with specific breast cancer subtypes, as seen in the multinomial logistic regression analyses discussed earlier.

The descriptive statistics confirm that the study sample consists primarily of older, overweight women—a profile consistent with typical breast cancer demographics. While the McNemar test results did not yield statistically significant associations after correction, they nonetheless contribute to a broader understanding of the genetic architecture of breast cancer by identifying trends suggestive of selective interdependencies among key DNA repair and checkpoint genes. These trends offer a basis for further investigation into combined genetic risk profiles and support the continued development of integrative models for breast cancer subtype stratification. Future studies with larger cohorts will be essential to validate these preliminary observations and explore their potential clinical relevance.

### 4.2. Multinomial Logistic Regression Analysis of Breast Cancer Subtypes

The multinomial logistic regression analysis in Table 5 explores the relationship between breast cancer histopathological subtypes—CDI LA, CDI LB, and CDI TNB—and a combination of genetic and clinical predictors. These subtypes were analyzed in three pairwise comparisons: CDI TNB vs. CDI LB, CDI LA vs. CDI LB, and CDI TNB vs. CDI LA. The genetic predictors evaluated were polymorphisms in XRCC1 (rs1799782), XPD (rs238406), and CHEK2 (rs17879961), while the clinical variables included BMI and age. The analysis sheds light on both statistically significant and non-significant associations, offering a nuanced understanding of how germline and phenotypic features may inform breast cancer classification.

#### 4.2.1. Distinct Genotypic and Phenotypic Profiles of CDI TNB vs. CDI LB

The comparison between CDI TNB and CDI LB yielded the strongest and most consistent pattern of statistically significant associations. All genetic and clinical predictors demonstrated *p*-values well below the Bonferroni-adjusted significance threshold (α = 0.0021), reinforcing the robustness of these findings. ORs and their 95% CIs supported the robustness of these findings, though many estimates reflected extreme values due to sparse data and potential quasi-complete separation. This indicates a clear distinction between these histopathological subtypes based on both genetic polymorphisms and clinical characteristics.

For the XRCC1 (rs1799782) gene, both genotype comparisons (AA vs. AT and TT vs. AT) showed highly significant associations with breast cancer subtype. The large negative estimates (−823.94 and −1212.59), along with near-zero ORs and 95% CI approaching zero, indicate that the AT genotype was markedly under-represented among TNB cases. This pattern suggests a potential protective role for the AT genotype in the context of TNB, possibly reflecting more efficient base excision repair (BER) in AT carriers that reduces the accumulation of oncogenic mutations. However, the low frequency of the TT genotype in the sample may have contributed to the extreme estimates observed, warranting cautious interpretation.

Variation in the XPD (rs238406) gene showed a strong genotype-dependent association with TNB. The AA vs. CA comparison revealed a positive association with TNB, while CC vs. CA was strongly negative, suggesting divergent roles in tumor subtype risk. These patterns imply that CA heterozygosity may support more balanced NER activity, while CC homozygosity could reflect impaired repair function, potentially increasing susceptibility to certain subtypes. The extreme ORs and wide or undefined 95% CI—particularly for the CC genotype—reflect the small subgroup sizes and warrant cautious interpretation. Nevertheless, these results underscore the potential importance of NER pathway variation in influencing breast cancer subtype development.

The CHEK2 (rs17879961) gene, which plays a critical role in cell cycle checkpoint regulation, demonstrated a strong association with the TNB subtype. Both CC–TC and TT–TC genotype comparisons yielded highly significant and positive estimates, with corresponding ORs approaching infinity. This suggests a marked over-representation of the CC and TT homozygous genotypes in TNB cases relative to LB, indicating elevated risk associated with either variant. These findings support the hypothesis that defects in checkpoint control mechanisms, as mediated by CHEK2 variation, contribute to the aggressive phenotype of TNB. However, the absence of certain genotypes in specific subgroups likely inflated effect estimates, pointing to the need for larger samples to validate the role of these rare but potentially high-risk alleles.

BMI showed a strong inverse association with CDI TNB, with a markedly negative estimate (−68.02) and an extremely low OR (≈2.89 × 10^−30^), indicating that individuals with lower body mass were significantly more likely to present with triple-negative tumors. This finding may reflect differences in metabolic or hormonal environments, particularly among postmenopausal women, where reduced adiposity is often linked to lower estrogen production but heightened systemic inflammation or immune dysregulation. These biological shifts are known to influence TNB biology and may interact with specific genetic profiles to modulate subtype risk. Although the precise mechanistic pathways remain to be fully elucidated, these data underscore the critical role of BMI in distinguishing TNB from other subtypes.

Age was positively associated with TNB status (estimate = 27.70; OR ≈ 1.07 × 10^12^), indicating that older individuals in this cohort were more likely to develop TNB. This finding is particularly noteworthy given that TNBC is typically more common in younger women. The divergence from established epidemiologic trends may be explained by population-specific factors, selection bias, or environmental exposures. It also raises the possibility that age-related processes—such as immunosenescence, declining DNA repair efficiency, and the accumulation of somatic mutations—may interact with the genetic variants studied to elevate susceptibility to the TNB subtype in older individuals.

Although age and BMI were included as independent covariates in the regression model, the possibility of synergistic interactions with genetic polymorphisms cannot be excluded. Larger studies with formal interaction analyses are needed to determine whether these clinical factors modify the effect of genetic variants on breast cancer subtype risk. Overall, the findings highlight a distinct clinical and molecular signature for TNB, shaped by both genotype-specific and phenotype-related features. While the direction of associations aligns with biological expectations, the extreme OR values and unstable 95% CIs—particularly for rare genotypes—underscore the impact of limited sample size and sparse data. Replication in larger, independent cohorts will be critical to confirm these associations and to further elucidate the interplay between genetic and clinical factors in breast cancer subtype development.

#### 4.2.2. Limited Genetic Differentiation Between Luminal Subtypes: CDI LA vs. CDI LB

In contrast to the robust associations observed for the TNB subtype, the comparison between LA and LB breast cancers yielded predominantly non-significant results with modest effect sizes across both genetic and clinical predictors. Most genotype-based comparisons failed to meet the Bonferroni-adjusted significance threshold (α = 0.0021), and 95% CIs frequently included 1, underscoring the limited discriminatory power of these markers between the two-hormone receptor–positive subtypes. Specifically, XRCC1 and XPD genotypes did not show significant associations with luminal subtype classification, with *p*-values exceeding 0.1 in all cases and effect sizes ranging narrowly from −1.82 to 1.03. These findings suggest that the genetic variants examined may have a minimal influence on the biological distinction between LA and LB tumors.

For the XRCC1 (rs1799782) gene, neither the AA–AT nor TT–AT genotype contrasts yielded meaningful ORs (0.259 and 3.18 × 10^−4^, respectively), with extremely wide 95% CIs (up to 211.28), reflecting a high degree of estimation uncertainty likely driven by sparse data and small sample sizes for the TT genotype. The broad CIs and lack of consistent directionality suggest that XRCC1 variants do not significantly contribute to the differentiation between LA and LB tumors in this cohort. This aligns with the known role of XRCC1 in base excision repair, a pathway more relevant to oxidative and alkylation damage, and perhaps less involved in the hormone-driven biology that characterizes luminal subtypes.

Similarly, XPD (rs238406) genotype comparisons showed no consistent or statistically significant effects. The ORs for AA–CA and CC–CA were 0.162 and 0.409, respectively, with wide CIs that span the null value. Although the CA genotype appeared more frequent overall, the absence of clear trends and the overlap in distributions across subtypes diminish the likelihood of a true subtype-specific role for this variant. Given XPD’s role in nucleotide excision repair, it remains plausible that its impact may be more pronounced in subtypes associated with genotoxic stress (e.g., TNB) rather than hormone-receptor–driven tumors.

CHEK2 (rs17879961) demonstrated a potentially noteworthy, though statistically non-significant, association with luminal subtype differentiation. The TT–TC genotype comparison produced a positive estimate (3.49) and an OR of 32.63 (95% CI: 1.85–574.63), with a nominal *p*-value of 0.017. Although this did not meet the Bonferroni-adjusted threshold for significance (α = 0.0021), it may suggest a biologically relevant trend. Given CHEK2’s established role in cell cycle checkpoint control and genomic stability, this elevated OR could point to an influence on the more proliferative LB phenotype, which is typically associated with higher tumor grade and Ki-67 expression. However, the wide 95% CI and low frequency of TT carriers necessitate cautious interpretation and underscore the need for validation in larger, independent cohorts.

Clinical predictors—BMI and age—did not show statistically significant contributions to distinguishing between LA and LB subtypes. Both variables had *p*-values well above the Bonferroni-adjusted threshold (0.695 for BMI and 0.985 for age), and their corresponding estimates were minimal (OR = 1.04, 95% CI: 0.86–1.26 for BMI; OR = 1.001, 95% CI: 0.89–1.13 for age), clustering closely around the null. These results indicate that neither body composition nor age served as meaningful discriminators between luminal subtypes in this cohort. This lack of association may reflect the biological and clinical similarities between LA and LB tumors, both of which share hormone receptor positivity. It is likely that more nuanced molecular differences—such as proliferation indices, HER2 signaling, or intrinsic gene expression profiles—underlie the classification of these subtypes and were not captured by the clinical variables included in this model.

Regarding the biological interplay between age, BMI, and luminal subtypes, both age and BMI were modeled as independent covariates without interaction terms. Although they did not exhibit synergistic effects with the genotypes analyzed, their inclusion remains essential for adjusting potential confounding. Future analyses incorporating interaction terms may clarify whether these clinical variables modulate the influence of genetic variants on subtype risk. Taken together, the results suggest that while the assessed genetic and clinical markers are informative for distinguishing biologically distinct subtypes—such as TNB versus luminal—they offer limited discriminatory power within the luminal category itself. The suggestive association observed for CHEK2 highlights a potentially meaningful avenue for further investigation, especially given its functional relevance. However, interpretation remains constrained by small sample sizes and wide 95% CIs. Future studies employing larger cohorts, more refined clinical data (e.g., Ki-67 proliferation index), and comprehensive molecular profiling may better resolve subtype-specific patterns within the luminal spectrum.

#### 4.2.3. Reaffirming Triple-Negative Distinctiveness: CDI TNB vs. CDI LA

This comparison mirrored the findings observed in the TNB vs. LB model, with all genetic and clinical predictors reaching significance levels below the Bonferroni-adjusted threshold (*p* < 0.0021), reaffirming the distinct molecular and clinical profile of triple-negative tumors. These consistently strong associations support the idea that TNB represents a biologically unique subtype within the breast cancer spectrum.

For XRCC1, the AA–AT and TT–AT genotypes showed large negative estimates (−598.78 and −907.53), indicating a markedly lower prevalence of AT genotype carriers in the TNB group used for this study. Correspondingly, the XRCC1 (rs1799782) polymorphism demonstrated substantially reduced odds for both contrasts (OR = 9.01 × 10^−261^ and OR = 0.0000, respectively), with extremely small values and wide or undefined 95% CIs. This consistent pattern reinforces the potential protective nature of the AT genotype against more aggressive tumor forms such as TNB. However, the breadth of the 95% CIs—particularly for the rare TT variant—also highlights the instability of these estimates, likely driven by sparse genotype counts. Despite these limitations, the results suggest that XRCC1 variants may influence subtype susceptibility, possibly through their role in base excision repair and maintenance of genomic stability.

A similarly striking pattern was observed for XPD (rs238406), where genotypic contrasts revealed significant divergence in subtype association. The AA–CA genotype exhibited a markedly elevated odds ratio (OR = 6.73 × 10^143^), while the CC–CA contrast produced an OR of 0.0000, reflecting an opposing effect and a strong depletion of the CC genotype in the TNB group. These findings align with the large positive and negative model estimates (331.18 for AA–CA and −880.68 for CC–CA, respectively), underscoring a robust genotypic gradient across subtypes. The magnitude of these associations, though accompanied by expansive 95% CIs (e.g., up to 2.08 × 10^147^), emphasizes the potential functional relevance of XPD in TNB. Biologically, this may be mediated through variation in NER efficiency, where differential DNA repair capacity could influence tumor development and subtype evolution in the context of genotoxic stress.

Among the most pronounced associations were those involving CHEK2 (rs17879961), where both the CC–TC and TT–TC genotype contrasts showed extremely elevated odds of triple-negative classification (OR = 6.69 × 10^229^ and OR = 3.81 × 10^217^, respectively), accompanied by wide but coherent 95% CIs. These striking results, reflected in the large positive model estimates (529.19 and 500.99), confirm the heightened risk associated with these genotypes and suggest that checkpoint failure mechanisms may play a pivotal role in the molecular etiology of triple-negative tumors. The complete or near-complete absence of certain genotypes in specific subtype groups likely contributed to the extremity of these estimates, reinforcing the idea that homozygosity at this locus—whether for the risk or alternate allele—substantially alters susceptibility. This aligns with the established biological role of CHEK2 in maintaining genomic integrity through cell cycle regulation, a function critically compromised in aggressive, genomically unstable tumor types such as TNB.

Both BMI and age emerged as statistically significant independent predictors of TNB when applying the Bonferroni-adjusted threshold (α = 0.0021). A strong negative association was observed for BMI (estimate = −54.16; OR = 3.00 × 10^−24^, *p* < 0.001), indicating that lower body mass index was linked to a markedly higher likelihood of TNB presentation. In contrast, age showed a strong positive relationship with TNB status (estimate = 20.49; OR = 7.95 × 10^8^, *p* < 0.001), suggesting that older individuals in this cohort were disproportionately represented among TNB cases. These trends—leaner body composition and older age—were consistent across models, with both predictors exhibiting 95% CIs that far exceeded conventional thresholds. While the association between lower BMI and TNB aligns with hypotheses related to hormonal environments and metabolic state, the positive effect of age contrasts with broader epidemiologic findings that typically link TNB to younger patients. This discrepancy may point to cohort-specific influences, such as genetic background, environmental exposures, or selection bias, and warrants further exploration in more diverse and larger datasets.

Together, these findings highlight a distinct clinical and molecular profile for TNB, characterized by extreme genotypic contrasts and independent contributions from BMI and age. The absence of interaction effects suggests that these clinical variables and genetic variants act through separate biological pathways—metabolic and hormonal for BMI and age, versus DNA repair and checkpoint control for the genetic markers. Despite limitations due to sample size and rare genotypes, the consistency and strength of these associations point to meaningful subtype-specific mechanisms that warrant further investigation in larger, more detailed studies.

#### 4.2.4. Interpretation and Translational Relevance

Taken together, the results underscore that the TNB subtype is genomically and clinically distinct from LA and LB forms. The consistent significance of DNA repair and checkpoint gene variants in TNB-related comparisons suggests a biologically meaningful disruption of genomic stability mechanisms. The absence of significant associations in the CDI LA vs. CDI LB model further supports the notion that luminal subtypes are more similar and possibly less driven by germline genetic variation in the DNA repair pathways assessed.

The extreme coefficient estimates (e.g., −1212.59 for XRCC1 and −963.02 for XPD) and very small *p*-values suggest a strong signal; however, they also raise the possibility of sparse data artifacts, especially if certain genotype combinations are rare. These results should be interpreted with caution, and larger sample sizes would help confirm these patterns and provide more stable effect estimates.

From a translational standpoint, these findings support the value of genotyping DNA repair polymorphisms as part of a risk stratification tool, particularly for TNB, a subtype characterized by poor prognosis and limited treatment options. If validated in broader cohorts, these genetic markers could inform targeted therapy decisions, especially in the context of PARP inhibitors and other agents exploiting DNA repair deficiencies.

This analysis offers compelling preliminary evidence for the involvement of specific DNA repair polymorphisms and clinical factors in breast cancer subtype differentiation. It highlights the potential of genetic profiling in refining diagnostic classifications and paving the way toward more personalized treatment strategies in breast cancer. Further research should aim to replicate these findings and explore the underlying molecular mechanisms through which these variants exert their influence on tumor biology.

### 4.3. Implications of DNA Repair Polymorphisms in Breast Cancer

DNA repair pathways like homologous recombination (HR) and base excision repair (BER) protect against mutations. Polymorphisms in repair genes (BRCA1, BRCA2, CHEK2, XRCC1, XPD) can reduce DNA repair efficiency, leading to genomic instability and increased breast cancer risk. Understanding these polymorphisms aids in risk pre-diction and development of targeted therapies iPARP [35,36]. In a previous study, we ob-served that the CHEK2 (rs17879961), was reported as a recurrent mutation accounting for a majority of (18) 72% of the pathogenic variants for this gene and 13% of all non-BRCA pathogenic variants in the cohort [37]. Although CHEK2 pathogenic variants are considered moderate-risk factors in breast cancer, the fact that it is so frequently found in breast cancer patients hypothesizes that there are other potentiating risk factors of the same molecular pathways or alternative pathways. Polygenic risk scores could bring more precision and risk estimation, especially in the case of mutations with moderate penetrance [38,39].

Important not only for risk assessment but also for treatment, identifying DNA repair actionable markers could anticipate forthcoming advancements in targeted therapy for cancer treatment [40]. Considering the typical association between DNA repair deficiencies and TNB histology known to be the most difficult form to treat and with the highest risk of relapse, it is even more important to identify molecular subtypes eligible for new therapies [41,42].

## 5. Future Directions and Implications

The findings of this preliminary study provide a compelling foundation for future research on the role of DNA repair gene polymorphisms in breast cancer heterogeneity. The distinct genotypic profiles associated with TNB suggest the need for further investigation into the mechanisms through which these genetic variants influence tumor development, progression, and therapeutic responsiveness. Building upon the observed associations, several avenues of inquiry are warranted.

First, replication of these findings in larger, multi-center cohorts is essential to confirm the statistical robustness of the associations and to mitigate the limitations imposed by the relatively small sample size of the present study. Larger sample sizes will also allow for more precise estimation of effect sizes and provide the statistical power needed to explore potential interactions between polymorphisms, environmental exposures, and clinical variables. Additionally, stratification by menopausal status, ethnicity, and family history could uncover subtype-specific patterns and population-level differences in genetic susceptibility.

Second, future studies should consider integrating functional assays to validate the biological relevance of the identified variants. For example, in vitro and in vivo studies examining DNA repair capacity, cell cycle checkpoint fidelity, and apoptosis regulation in cells carrying the AT variant of XRCC1 or the TT variant of CHEK2 could elucidate the mechanistic pathways through which these polymorphisms confer risk or protection. Moreover, transcriptomic and proteomic profiling of tumors stratified by genotype may reveal downstream effectors and pathways that are differentially regulated, offering insights into targetable vulnerabilities.

Third, the clinical implications of these findings must be further explored in the context of personalized oncology. The strong association between certain genotypes and the TNB phenotype—a subtype notoriously lacking targeted treatment options—suggests the potential utility of incorporating germline genetic profiling into early diagnostic and prognostic workflows. For example, patients with high-risk variants may benefit from enhanced screening protocols or early intervention strategies. More importantly, these variants may guide treatment selection, particularly in relation to emerging therapies such as PARP inhibitors or immune checkpoint blockade, which exploit DNA repair deficiencies [41].

Finally, the observed interdependency between some of the polymorphisms, particularly between XPD and both XRCC1 and CHEK2, raises important questions about polygenic interactions in tumorigenesis. Future analytic models should therefore adopt multi-locus approaches that account for gene–gene interactions and cumulative risk scores, rather than evaluating each polymorphism in isolation. This integrative strategy could improve subtype classification models and refine risk stratification tools, ultimately enhancing clinical decision-making [42].

While this study offers valuable early insights into the genotypic architecture of breast cancer subtypes, it also highlights the complexity of translating genetic data into clinical practice. Continued research, guided by mechanistic validation, larger patient cohorts, and clinical applicability, will be critical to advancing the field of precision oncology and improving outcomes for patients with breast cancer.

## 6. Limitations

As a preliminary investigation, this study offers important exploratory insights into the relationship between DNA repair gene polymorphisms and breast cancer subtypes. However, several limitations that may influence the scope and interpretation of the findings must be acknowledged:Small Sample Size (N = 36): This limits the power to detect subtle associations and increases the likelihood of statistical errors, especially for rare genotypes or subgroup analyses.Limited Population Diversity: This study includes only female breast cancer patients from a single geographic region, which may limit applicability to broader populations with different genetic or environmental backgrounds.Risk of Overfitting: With a small sample and uneven genotype frequencies, the statistical models (e.g., logistic regression) may overfit the data, leading to inflated or misleading associations.Cross-Sectional Design: Data were collected at a single time point, preventing the analysis of cause-and-effect relationships between genotypes and clinical outcomes.Narrow Genetic Scope: Only three SNPs (XRCC1, CHEK2, XPD) were studied, which captures only a small part of the genetic landscape involved in breast cancer.No Functional Validation: This study relies on statistical associations without experimental data (e.g., gene expression or protein function) to support biological relevance.Incomplete Clinical Data: Key clinical variables such as hormonal status, treatment response, and family history were not included, as these details were not consistently available in the records, thereby limiting the depth of clinical interpretation.No Healthy Control Group: The absence of a cancer-free comparison group limits this study to within-case analyses, not risk prediction.

Taken together, these limitations are characteristic of preliminary research and are not unexpected at this stage. They underscore the importance of future studies involving larger, more diverse populations, comprehensive clinical data, and functional analyses to confirm and expand upon these initial findings.

## 7. Conclusions

This study provides preliminary evidence that specific DNA repair gene polymorphisms—XRCC1 (rs1799782), CHEK2 (rs17879961), and XPD (rs238406)—are significantly associated with the CDI TNB subtype. These associations suggest a potential role of impaired base excision, nucleotide excision, and checkpoint repair mechanisms in the development of this aggressive form of breast cancer. Notably, the AT genotype of XRCC1 and the CC and TT genotypes of CHEK2 were under-represented in the TNB group, indicating possible protective effects.

In contrast, few significant associations were found between these polymorphisms and the luminal subtypes (CDI LA vs. CDI LB), pointing to a greater genetic overlap between LA and LB tumors. Additionally, the genotype dependencies observed between XPD and both XRCC1 and CHEK2 suggest functional interactions that may influence tumorigenesis.

Despite its small sample size, this study supports the hypothesis that DNA repair polymorphisms can inform breast cancer subtype classification. These findings highlight the need for larger, multi-center studies with functional validation to confirm the clinical utility of these genetic markers in risk stratification and targeted therapy development, especially for TNB.

## Figures and Tables

**Figure 1 jcm-14-03764-f001:**
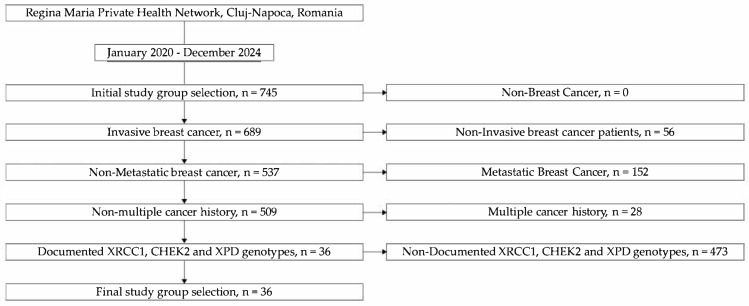
Patient selection flowchart. (Note: n = number of patients).

**Table 1 jcm-14-03764-t001:** Descriptive statistical analysis.

	Mean	SE	95% CI		Median	SD	Minimum	Maximum	Shapiro–Wilk	
			Lower	Upper					W	*p*
Age	62.1	1.84	58.4	65.9	63	11.04	38	82	0.98	0.735
BMI	28.7	1.02	26.6	30.7	28.5	6.1	18.2	46.6	0.97	0.418

**Table 2 jcm-14-03764-t002:** Contingency table between CHEK2 (rs17879961) and XRCC1 (rs.1799782) polymorphisms.

XRCC1 (rs.1799782)	CHEK2 (rs17879961)			Total
	TC	CC	TT	
AT	4	5	3	12
AA	8	8	4	20
TT	2	0	2	4
Total	14	13	9	36

**Table 3 jcm-14-03764-t003:** Contingency table between XPD (rs238406) and XRCC1 (rs.1799782) polymorphisms.

XRCC1 (rs.1799782)	XPD (rs238406)			Total
	CA	AA	CC	
AT	9	2	1	12
AA	8	7	5	20
TT	2	0	2	4
Total	19	9	8	36

**Table 4 jcm-14-03764-t004:** Contingency table between XPD (rs238406) and CHEK2 (rs17879961) polymorphisms.

XPD (rs238406)	CHEK2 (rs17879961)			Total
	TC	CC	TT	
CA	6	9	4	19
AA	2	4	3	9
CC	6	0	2	8
Total	14	13	9	36

**Table 5 jcm-14-03764-t005:** Multinomial logistic regression model coefficients for histopathological subtypes of breast cancer.

HP	Predictor	Estimate	SE	Z	*p*	OR	95% CI	
							Lower	Upper
CDI TNB–CDI LB	Intercept	−512.39147	2.624	−195.2720	<0.001	2.96e−223	1.73 × 10^−225^	5.07 × 10^−221^
	XRCC1 (rs.1799782):							
	AA–AT	−823.94469	1.7262	−477.3045	<0.001	0.000	0.0000	0.00
	TT–AT	−1212.58863	5.85 × 10^−13^	−2.07 × 10^−15^	<0.001	0.000	0.0000	0.00
	XPD (rs238406):							
	AA–CA	451.74345	1.7262	261.6913	<0.001	1.55 × 10^196^	5.25 × 10^194^	4.56 × 10^197^
	CC–CA	−963.01555	7.92 × 10^−14^	−1.22 × 10^−16^	<0.001	0.000	NaN	NaN
	CHEK2 (rs17879961):							
	CC–TC	784.00225	11.7413	66.7730	<0.001	Inf	Inf	Inf
	TT–TC	777.40846	14.3553	54.1548	<0.001	Inf	Inf	Inf
	BMI	−68.01547	4.4690	−15.2194	<0.001	2.89 × 10^−60^	4.54 × 10^−68^	1.84 × 10^−52^
	Age	27.70229	1.8797	14.7376	<0.001	1.07 × 10^24^	2.70 × 10^20^	4.28 × 10^26^
CDI LA–CDI LB	Intercept	−1.34957	3.4198	−0.3946	0.693	0.259	3.18 × 10^−12^	211.28
	XRCC1 (rs.1799782):							
	AA–AT	−1.82255	1.2414	−1.4682	0.142	0.162	0.0142	1.84
	TT–AT	−0.89443	1.7996	−0.4970	0.619	0.409	0.0120	13.91
	XPD (rs238406):							
	AA–CA	1.03707	1.2343	0.8402	0.401	2.821	0.2511	31.70
	CC–CA	−1.06826	1.6772	−0.6369	0.524	0.344	0.0128	9.20
	CHEK2 (rs17879961):							
	CC–TC	0.18311	1.2977	0.1411	0.888	1.201	0.0944	15.28
	TT–TC	3.48527	1.4635	2.3814	0.017	32.631	1.8530	574.63
	BMI	0.03907	0.0998	0.3916	0.695	1.040	0.8552	1.26
	Age	0.00114	0.0599	0.0191	0.985	1.001	0.8902	1.13
CDI TNB–CDI LA	Intercept	−217.26075	6.2112	−34.9788	<0.001	4.41 × 10^−190^	2.28 × 10^−100^	8.55 × 10^−180^
	XRCC1 (rs.1799782):							
	AA–AT	−598.77619	4.1008	−146.0137	<0.001	9.01 × 10^−261^	2.91 × 10^−264^	2.79 × 10^−257^
	TT–AT	−907.53428	2.64 × 10^−24^	−3.43 × 10^−26^	<0.001	0.0000	NaN	NaN
	XPD (rs238406):							
	AA–CA	331.17660	4.1008	80.7586	<0.001	6.73 × 10^143^	2.18 × 10^140^	2.08 × 10^147^
	CC–CA	−880.68313	0.0000	−Inf	<0.001	0.0000	0.00000	0.000
	CHEK2 (rs17879961):							
	CC–TC	529.19332	27.9455	18.9366	<0.001	6.69 × 10^229^	1.09 × 10^206^	4.10 × 10^253^
	TT–TC	500.99770	34.1327	14.6779	<0.001	3.81 × 10^217^	3.36 × 10^188^	4.31 × 10^246^
	BMI	−54.16329	10.6606	−5.0807	<0.001	3.00 × 10^−48^	2.53 × 10^−66^	3.56 × 10^−30^
	Age	20.49428	4.4769	4.5778	<0.001	7.95 × 10^24^	122972.76617	5.14 × 10^24^

## Data Availability

Data are contained within the article (see Appendix A).

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
