# Peer review of "Associations Between DNA Repair Gene Polymorphisms and Breast Cancer Histopathological Subtypes: A Preliminary Study"

_jcm, 2025, doi:10.3390/jcm14113764_

Round 1
Reviewer 1 Report
Comments and Suggestions for Authors
This study presents preliminary evidence indicating that specific DNA repair gene polymorphisms—XRCC1 (rs1799782), CHEK2 (rs17879961), and XPD (rs238406)—are significantly associated with the triple-negative breast cancer (TNBC) subtype. These findings suggest a potential role for impaired base excision repair, nucleotide excision repair, and checkpoint repair mechanisms in the development of this aggressive form of breast cancer. This research provides valuable early insights into the genotypic architecture underlying breast cancer subtypes and supports further investigation into the genetic basis of TNBC. The topic aligns well with the scope of the Journal and is worthy of publication following minor revisions.
- In the discussion section, emphasis should be placed on analyzing the reasons behind the correlation between different parameters, rather than merely reiterating the range of variation for the research parameters mentioned in the results section. Specifically, for lines 241-254, it is crucial to delve into why indicators such as BMI can significantly predict subtypes of breast cancer. Furthermore, why the TNB group showed a lower prevalence of AT genotype (Lines 370-373)?
- The format of the references list must adhere to the specific requirements outlined by the Journal.
Author Response
1. Summary
Thank you very much for taking the time to review this manuscript. Please find the detailed responses below and the corresponding revisions and corrections highlighted in the re-submitted files.
2. Point-by-point response to Comments and Suggestions for Authors
Comment 1. This study presents preliminary evidence indicating that specific DNA repair gene polymorphisms—XRCC1 (rs1799782), CHEK2 (rs17879961), and XPD (rs238406)—are significantly associated with the triple-negative breast cancer (TNBC) subtype. These findings suggest a potential role for impaired base excision repair, nucleotide excision repair, and checkpoint repair mechanisms in the development of this aggressive form of breast cancer. This research provides valuable early insights into the genotypic architecture underlying breast cancer subtypes and supports further investigation into the genetic basis of TNBC. The topic aligns well with the scope of the Journal and is worthy of publication following minor revisions.
Answer 1. We sincerely thank the reviewer for the kind and supportive words regarding our manuscript. We greatly appreciate your positive evaluation and fully agree with your observations and suggestions. We have implemented the recommended revisions, and we will provide detailed explanations of these changes in our responses to the specific comments that follow.
Comment 2. In the discussion section, emphasis should be placed on analyzing the reasons behind the correlation between different parameters, rather than merely reiterating the range of variation for the research parameters mentioned in the results section. Specifically, for lines 241-254, it is crucial to delve into why indicators such as BMI can significantly predict subtypes of breast cancer. Furthermore, why the TNB group showed a lower prevalence of AT genotype (Lines 370-373)?
Answer 2. We would like to sincerely thank the reviewer for their insightful comments and valuable suggestions. We fully agree with the points raised regarding the need to deepen the analysis in the discussion section rather than simply reiterating the findings.
In response, we have revised subsections 4.2.1 (lines 407-473), 4.2.2 (lines 476-538), and 4.2.3 (lines 541-603) of the Discussion to provide a more meaningful interpretation of the results. These revisions emphasize the reasoning behind the observed correlations between the variables analyzed in our study. Specifically, we have expanded on the role and significance of BMI and Age as predictive indicators for different breast cancer subtypes, offering a more in-depth interpretation of their relevance within the clinical and biological context (lines 444-462 / 514-524 / 582-595).
Furthermore, we addressed the lower prevalence of the AT genotype observed in the TNB group (former lines 370–373, now lines 546-548). We clarified that this finding reflects a characteristic specific to the sample population on which the study was based, and we have now explained this more clearly in the revised text.
We appreciate the reviewer’s input, which has helped us improve the clarity and depth of our discussion.
Comment 3. The format of the references list must adhere to the specific requirements outlined by the Journal.
Answer 3. We thank the reviewer for pointing this out. We have thoroughly revised the reference list and reformatted all references to ensure full alignment with the guidelines of the Journal of Clinical Medicine.
Reviewer 2 Report
Comments and Suggestions for Authors
This research aims to assess whether there is a statistical association or co-occurrence between specific polymorphisms in individuals and evaluate their predictive relevance for differentiating among major histopathological classifications. The authors successfully achieved their goal by demonstrating that DNA repair gene polymorphisms (rs1799782) and XPD (rs238406) are significantly associated with the triple-negative breast cancer (CDI TNB) subtype.
However, I suggest that the paper should provide more background in the introduction, for example:
- Explain the different subtypes of breast cancer and their characteristic.
- The role of the DNA repair pathway in the development of breast cancer.
- The contribution of XRCC1 and XPD in breast cancer progression.
Finally, there is a typo in line 111; the sentence "no evidence of metastatic disease at the time of diagnosis" is repeated.
Author Response
1. Summary
Thank you very much for taking the time to review this manuscript. Please find the detailed responses below and the corresponding revisions and corrections highlighted in the re-submitted files.
2. Point-by-point response to Comments and Suggestions for Authors
Comment 1. This research aims to assess whether there is a statistical association or co-occurrence between specific polymorphisms in individuals and evaluate their predictive relevance for differentiating among major histopathological classifications. The authors successfully achieved their goal by demonstrating that DNA repair gene polymorphisms (rs1799782) and XPD (rs238406) are significantly associated with the triple-negative breast cancer (CDI TNB) subtype.
Answer 1. We thank the reviewer for the positive and encouraging feedback. We are pleased to hear that our research objectives and findings were clearly conveyed.
Comment 2. However, I suggest that the paper should provide more background in the introduction, for example:
- Explain the different subtypes of breast cancer and their characteristic.
- The role of the DNA repair pathway in the development of breast cancer.
- The contribution of XRCC1 and XPD in breast cancer progression.
Answer 2. We sincerely thank the reviewer for their insightful suggestions, which have significantly contributed to improving the quality and clarity of our manuscript. Incorporating this feedback has helped us refine the introduction and provide a more comprehensive background, enhancing the manuscript’s scientific depth and accessibility.
In response to the first suggestion—“Explain the different subtypes of breast cancer and their characteristics”—we have revised the Introduction section to include a new subsection titled 1.1. Breast Cancer Histopathological Types(lines 55–79), where we discuss the main subtypes of breast cancer, their defining histopathological features, and clinical relevance.
For the second and third suggestions—“The role of the DNA repair pathway in the development of breast cancer”and “The contribution of XRCC1 and XPD in breast cancer progression”—we have added a dedicated subsection titled 1.2. DNA Repair Pathways in Breast Cancer: Roles of XRCC1, XPD, and CHEK2 (lines 81-133). In this section, we elaborate on the significance of DNA repair mechanisms in breast cancer, specifically focusing on the nucleotide excision repair (NER) and base excision repair (BER) pathways. Furthermore, we detail the molecular and pathophysiological roles of XRCC1, XPD, and CHEK2, highlighting their relevance to breast cancer progression and genomic stability.
We believe these additions enhance the contextual foundation of our study and help position our findings within the broader landscape of breast cancer research. Thank you again for helping us move the manuscript toward its best form.
Comment 3. Finally, there is a typo in line 111; the sentence "no evidence of metastatic disease at the time of diagnosis" is repeated.
Answer 3. We thank the reviewer for drawing our attention to this oversight. We have removed the repeated inclusion criterion in line 111 to correct the duplication.
Reviewer 3 Report
Comments and Suggestions for Authors
This study explores the distribution and interactions of three polymorphisms—XRCC1 (rs1799782), CHEK2 (rs17879961), and XPD (rs238406)—in Romanian breast cancer patients, with a focus on their associations with histopathological subtypes, age, and BMI. The research addresses an important gap in understanding the genetic underpinnings of breast cancer, particularly triple-negative breast cancer (TNB), and provides preliminary evidence for the role of DNA repair gene variants in disease susceptibility. While the findings are intriguing, some methodological and interpretive aspects warrant clarification to strengthen the study’s impact.
1. Sample Size: The small cohort (n = 36) limits statistical power, particularly for subgroup analyses (e.g., luminal vs. TNB comparisons). A larger sample would improve the reliability of associations.
2. Population Specificity: The findings may not be generalizable due to the single-center, Romanian-only cohort. Including multi-center or multi-ethnic data would enhance external validity.
3. Correction for Multiple Testing: The study reports multiple statistical comparisons (e.g., McNemar’s tests, regression models) but does not mention adjustments for multiple testing (e.g., Bonferroni correction). This increases the risk of Type I errors.
4. Missing Functional Data: While statistical associations are reported, mechanistic insights (e.g., how these SNPs affect protein function or DNA repair efficiency) are lacking. Incorporating functional studies or referencing prior functional evidence would strengthen conclusions.
6. BMI and Age Analysis: The inclusion of BMI and age in the regression model is useful, but their biological interaction with genetic variants remains speculative. Clarifying whether these are independent or synergistic effects would be valuable.
7. Confounding Factors: Were other potential confounders (e.g., family history, hormonal status, lifestyle factors) considered in the analysis?
8. Effect Sizes: Reporting odds ratios (ORs) and confidence intervals (CIs) for significant associations would help assess clinical relevance.
9. Reproducibility: Were genotyping methods validated (e.g., via Sanger sequencing or replication in a separate cohort)?
10. Please unify the format of references in the article, including the author's name, the case of words in the title of the article, the writing of the name of the journal, and the page number.
Comments on the Quality of English LanguageThe English could be improved to more clearly express the research.
Author Response
1. Summary
Thank you very much for taking the time to review this manuscript. Please find the detailed responses below and the corresponding revisions and corrections highlighted in the re-submitted files.
2. Point-by-point response to Comments and Suggestions for Authors
Comment 1. This study explores the distribution and interactions of three polymorphisms—XRCC1 (rs1799782), CHEK2 (rs17879961), and XPD (rs238406)—in Romanian breast cancer patients, with a focus on their associations with histopathological subtypes, age, and BMI. The research addresses an important gap in understanding the genetic underpinnings of breast cancer, particularly triple-negative breast cancer (TNB), and provides preliminary evidence for the role of DNA repair gene variants in disease susceptibility. While the findings are intriguing, some methodological and interpretive aspects warrant clarification to strengthen the study’s impact.
Answer 1. We thank the reviewer for recognizing the novelty and significance of our study, particularly in highlighting the relevance of XRCC1, CHEK2, and XPD polymorphisms in the context of breast cancer subtypes, including triple-negative breast cancer. We appreciate the acknowledgment of our contribution to understanding the genetic factors underlying breast cancer susceptibility in the Romanian population.
Regarding the reviewer’s comment on the need for clarification of certain methodological and interpretive aspects, we fully agree and are committed to addressing each point thoroughly. We provide detailed responses and corresponding revisions in the following answers to ensure our study’s rigor and clarity are enhanced.
Comment 2. Sample Size: The small cohort (n = 36) limits statistical power, particularly for subgroup analyses (e.g., luminal vs. TNB comparisons). A larger sample would improve the reliability of associations.
Answer 2. We thank the reviewer for their comment regarding the sample size. We fully acknowledge that the relatively small cohort (n = 36) limits the statistical power, particularly in subgroup analyses such as comparisons between luminal and triple-negative breast cancer cases. This limitation is explicitly addressed in the Limitations section of our manuscript (lines 701–703), where we note the challenges associated with deriving definitive conclusions from a small sample.
As mentioned in the Introduction (lines 151–155), the primary aim of our study was to conduct a preliminary, retrospective, and exploratory analysis of the distribution and potential interactions of XRCC1, CHEK2, and XPD polymorphisms in Romanian breast cancer patients. This work is intended to be hypothesis-generating, providing foundational insights that may guide the design of future studies with larger, prospective cohorts.
Due to the retrospective nature of this study and the fact that our analysis is based solely on existing patient records, expanding the cohort within the current study design was not feasible. Nonetheless, we believe the data presented here offer valuable early evidence, particularly in the underrepresented Romanian population, and contribute meaningfully to the broader efforts of integrating moderate-risk genetic variants into breast cancer risk assessment and molecular stratification strategies.
We appreciate the reviewer’s point and hope that our transparent acknowledgment of this limitation, along with the stated exploratory scope, will clarify the intent and contribution of our work within its methodological context.
Comment 3. Population Specificity: The findings may not be generalizable due to the single-center, Romanian-only cohort. Including multi-center or multi-ethnic data would enhance external validity.
Answer 3. We thank the reviewer for raising this important point regarding the generalizability of our findings. We fully recognize that the single-center, Romanian-only cohort represents a limitation in terms of external validity, and we have addressed this in the Limitations section of the manuscript (lines 704–706).
However, as stated in the Introduction (lines 151–155), the primary objective of our study was to conduct an exploratory, hypothesis-generating analysis focused on a Romanian breast cancer cohort—a population that remains underrepresented in current genetic and molecular oncology literature. While our findings may not be broadly generalizable at this stage, we believe they provide valuable population-specific insights that could inform the design of future multi-center or multi-ethnic studies.
Moreover, such localized data are essential for the advancement of precision oncology, as genetic risk factors and polymorphism distributions often vary significantly across ethnicities and geographic regions. By highlighting potential associations in a defined cohort, this study contributes to a more nuanced understanding of breast cancer genetics, which is critical for developing regionally tailored prevention and treatment strategies.
We appreciate the reviewer’s comment and agree that expanding future studies to include diverse populations and multiple centers will be important for validating and extending our findings.
Comment 4. Correction for Multiple Testing: The study reports multiple statistical comparisons (e.g., McNemar’s tests, regression models) but does not mention adjustments for multiple testing (e.g., Bonferroni correction). This increases the risk of Type I errors.
Answer 4. We thank the reviewer for this insightful and important observation regarding the need to adjust for multiple statistical comparisons, which is essential for minimizing the risk of Type I errors.
In response, we have now performed Bonferroni corrections for both the McNemar tests and the multinomial logistic regression models, as detailed in the revised manuscript (Lines 240–244 and 280–285, respectively).
As a result of this adjustment, the previously significant McNemar test results for genotype associations between XPD (rs238406) and XRCC1 (rs1799782) (Lines 258–260) and XPD (rs238406) and CHEK2 (rs17879961) (Lines 268–270) are no longer statistically significant under the Bonferroni-adjusted threshold (α = 0.0167). Accordingly, we have made the necessary corrections in the interpretation of these results in Section 4.1. Genetic and Clinical Characteristics of the Study Cohort and Interpolymorphism Associations, specifically in Lines 333–334, 345–349, and 359–363.
To ensure consistency, we have also revised the Abstract (Lines 30–32) to accurately reflect the updated findings and maintain alignment with the corrected statistical interpretations.
With regard to the multinomial logistic regression models, the application of the Bonferroni correction (α = 0.0021) did not alter the significance or interpretation of those results.
We are grateful for the reviewer’s attention to this methodological detail, which has helped us strengthen the statistical rigor and reliability of our analysis.
Comment 5. Missing Functional Data: While statistical associations are reported, mechanistic insights (e.g., how these SNPs affect protein function or DNA repair efficiency) are lacking. Incorporating functional studies or referencing prior functional evidence would strengthen conclusions.
Answer 5. We thank the reviewer for this thoughtful and valuable comment regarding the lack of functional data. We fully agree that integrating mechanistic insights is essential for strengthening the biological relevance of statistical associations.
In response to this suggestion, we have introduced a new subsection in the Introduction—1.2. DNA Repair Pathways in Breast Cancer: Roles of XRCC1, XPD, and CHEK2 (Lines 81–133)—to provide a detailed overview of the physiopathological roles of the three genes investigated in this study. This section discusses how each gene contributes to key DNA repair pathways (NER and BER), and how their dysfunction may influence breast cancer development and subtype differentiation.
Additionally, we have integrated functional context and interpretation throughout the Results and Discussion sections, specifically in: Subsection 4.2.1 (Lines 406–473), Subsection 4.2.2 (Lines 475–538), and Subsection 4.2.3 (Lines 540–604)
For example, we included statements such as:
-
“This aligns with the known role of XRCC1 in base excision repair, a pathway more relevant to oxidative and alkylation damage, and perhaps less involved in the hormone-driven biology that characterizes luminal subtypes.”(Lines 492–495)
-
“Given XPD’s role in nucleotide excision repair, it remains plausible that its impact may be more pronounced in subtypes associated with genotoxic stress (e.g., TNB) rather than hormone-receptor–driven tumors.” (Lines 500–503)
-
“These striking results, reflected in the large positive model estimates (529.19 and 500.99), confirm the heightened risk associated with these genotypes and suggest that checkpoint failure mechanisms may play a pivotal role in the molecular etiology of triple-negative tumors.” (Lines 573–576)
These additions aim to ground our statistical findings in biological plausibility, drawing on existing knowledge of DNA repair pathways and their roles in breast cancer pathogenesis.
However, as the reviewer rightly notes, direct functional validation (e.g., assessing protein activity or DNA repair efficiency in vitro) was beyond the scope of this study, primarily due to its retrospective design and exclusive reliance on archived clinical and genetic data. The absence of fresh biological material precluded us from conducting additional laboratory-based assays.
We appreciate the reviewer’s suggestion, which has helped us enhance the interpretative depth of our work. We also emphasize that this study is intended as preliminary and hypothesis-generating, with the goal of laying the groundwork for future functional and translational research.
Comment 6. BMI and Age Analysis: The inclusion of BMI and age in the regression model is useful, but their biological interaction with genetic variants remains speculative. Clarifying whether these are independent or synergistic effects would be valuable.
Answer 6. We thank the reviewer for this insightful comment. We appreciate the suggestion to clarify the relationships between BMI, age, and the genetic variants investigated in our study. We agree that improving the explanation of these factors is important to enhance the clarity and interpretability of our results for readers.
In response, we have revised the Discussion section, specifically subsections 4.2.1, 4.2.2, and 4.2.3, to more clearly describe the observed associations for each pairwise comparison among LA, LB, and TNB. In these revised sections, we have detailed how BMI and age relate to the histopathological subtypes analyzed and how they may act as independent or potentially interacting factors in the context of our findings (CDI TNB vs. CDI LB [lines 444–473], CDI LA vs. CDI LB [ lines 514–538], CDI TNB vs. CDI LA [lines 583–604]).
We hope these clarifications adequately address the reviewer’s concern and contribute to a more comprehensive understanding of the results.
Comment 7. Confounding Factors: Were other potential confounders (e.g., family history, hormonal status, lifestyle factors) considered in the analysis?
Answer 7. We thank the reviewer for this thoughtful and important comment. We fully agree that the inclusion of additional potential confounding factors—such as family history, hormonal status, and lifestyle-related variables—would have strengthened the clinical interpretation and the robustness of our analysis.
Indeed, we aimed to include these variables; however, due to the retrospective nature of the study and the limitations inherent to the available medical records, these factors were not consistently or systematically recorded for all patients. As such, they could not be reliably included in our analysis without introducing substantial bias or missing data issues.
The only confounding factors that were consistently documented and thus suitable for inclusion in the regression models were age and BMI. These were incorporated into the analysis accordingly.
We have acknowledged this as a limitation in Section 6. Limitations, lines 719–722, where we explicitly state the constraints related to missing clinical and lifestyle information. We hope the reviewer understands these limitations were not due to study design choices but rather to the quality and completeness of the existing records in this retrospective cohort.
We believe this clarification enhances the transparency of our study and we appreciate the reviewer’s comment for highlighting the importance of addressing potential confounders.
Comment 8. Effect Sizes: Reporting odds ratios (ORs) and confidence intervals (CIs) for significant associations would help assess clinical relevance.
Answer 8. We sincerely thank the reviewer for this valuable suggestion. We fully agree that including effect sizes in the form of ORs along with their 95% CIs greatly enhances the interpretability and clinical relevance of the findings.
In response to this comment, we have updated Table 5 (line 292) to include three new columns reporting the ORs, 95% CI lower bound, and 95% CI upper bound for each significant association identified in the multinomial logistic regression analysis.
Furthermore, we have integrated the interpretation of these effect sizes into the Discussion section to provide a clearer understanding of their implications. Specifically, the results have been discussed in subsections: 4.2.1 (lines 405–472), 4.2.2 (lines 474–537), 4.2.3 (lines 539–603)
We believe these additions significantly strengthen the manuscript by providing a clearer picture of the magnitude and precision of associations, in line with the reviewer’s recommendation. We are grateful for this constructive feedback, which helped improve the clarity and rigor of our study.
Comment 9. Reproducibility: Were genotyping methods validated (e.g., via Sanger sequencing or replication in a separate cohort)?
Answer 9. We thank the reviewer for this important comment highlighting the need for validating the reproducibility of our genotyping results. We fully agree that ensuring the accuracy and reliability of genetic findings is essential, particularly in studies involving associations with clinical outcomes.
To address this, we confirm that Sanger sequencing—widely regarded as the gold standard for DNA sequence validation—was used to confirm all significant genotyping findings. Specifically, validation was performed using the ProDye® Terminator Sequencing System (Promega Corporation, Madison, WI, USA), as well as through the professional sequencing services of Eurofins Genomics (Ebersberg, Germany). Additionally, XRCC1 (rs1799782) and XPD (rs238406) were genotyped using the multiplex PCR-restriction fragment length polymorphism (M-PCR-RFLP) technique, as detailed in references [References 33 &34]. (Lines 192-198)
Regarding replication in an independent dataset or cohort, we acknowledge that this was not performed in the present study. The primary reason is the retrospective design and the limited availability of comparable external datasets with both clinical and genetic data matching our inclusion criteria. Moreover, the nature of the patient population and the reliance on archived samples within a single center restricted the feasibility of external replication.
Once again, we thank the reviewer for this helpful suggestion, which has allowed us to clarify the methodological rigor and limitations of our work.
Comment 10. Please unify the format of references in the article, including the author's name, the case of words in the title of the article, the writing of the name of the journal, and the page number.
Answer 10. We thank the reviewer for bringing this oversight to our attention. We sincerely appreciate this comment, as consistent and properly formatted references are essential for maintaining the professionalism and readability of the manuscript.
In response, we have carefully reviewed and revised all references to ensure they are fully aligned with the Journal of Clinical Medicine’s formatting guidelines. This includes unifying the presentation of authors’ names, standardizing the use of title case for article titles where required, formatting journal names appropriately (using correct abbreviations and styles), and ensuring that page numbers and other citation details are complete and consistent throughout the reference list.
We are grateful for the reviewer’s attention to detail, which has helped us improve the quality and presentation of the manuscript.
Comment 11. The English could be improved to more clearly express the research.
Answer 11. We thank the reviewer for this valuable observation. In response, we have carefully revised the entire manuscript to improve the clarity, flow, and precision of the language. Our goal was to ensure that the phrasing accurately reflects the scientific content and that the writing clearly conveys the key messages of the study.
We paid particular attention to sentence structure, word choice, and consistency of terminology, aiming to make the text more accessible and easier to follow for readers. These revisions were made to help readers better understand the methodology, interpret the results, and grasp the main conclusions of the study without ambiguity.
We sincerely appreciate the reviewer’s comment, which helped us enhance the overall quality and readability of the manuscript.
Round 2
Reviewer 3 Report
Comments and Suggestions for Authors
The authors have addressed all my concerns, I rocmmend accept it in current form.